# Trilobatin, a Novel SGLT1/2 Inhibitor, Selectively Induces the Proliferation of Human Hepatoblastoma Cells

**DOI:** 10.3390/molecules24183390

**Published:** 2019-09-18

**Authors:** Lujing Wang, Min Liu, Fei Yin, Yuanqiang Wang, Xingan Li, Yucui Wu, Cuilian Ye, Jianhui Liu

**Affiliations:** Chongqing Key Lab of Medicinal Chemistry & Molecular Pharmacology, Chongqing University of Technology, Chongqing 400054, China; Wanglujinggot7@163.com (L.W.); 18875083212@163.com (M.L.); wangyqnn@cqut.edu.cn (Y.W.); lixingan@foxmail.com (X.L.); cqwuyucui@163.com (Y.W.); ycl@cqut.edu.cn (C.Y.)

**Keywords:** Anti-tumor, Na^+^-d-glucose co-transporter (SGLT) inhibitors, Cell proliferation, Trilobatin

## Abstract

Studies have indicated that Na^+^-d-glucose co-transporter (SGLT) inhibitors had anti-proliferative activity by attenuating the uptake of glucose in several tumor cell lines. In this study, the molecular docking showed that, trilobatin, one of the dihydrochalcones from leaves of *Lithocarpus polystachyus* Rehd., might be a novel inhibitor of SGLT1 and SGLT2, which evidently attenuated the uptake of glucose in vitro and in vivo. To our surprise, we observed that trilobatin did not inhibit, but promoted the proliferation of human hepatoblastoma HepG2 and Huh 7 cells when it was present at high concentrations. At the same time, incubation with high concentrations of trilobatin arrested the cell cycle at S phase in HepG2 cells. We also found that treatment with trilobatin had no significant effect on the expression of hepatitis B x-interacting protein (HBXIP) and hepatocyte nuclear factor (HNF)-4α, the two key regulators of hepatocyte proliferation. Taken together, although trilobatin worked as a novel inhibitor of SGLTs to attenuate the uptake of glucose, it also selectively induced the cell proliferation of HepG2 cells, suggesting that not all the SGLT inhibitors inhibited the proliferation of tumor cells, and further studies are needed to assess the anti-cancer potentials of new glucose-lowering agents.

## 1. Introduction 

A large body of evidence showed that tumor cells were easy to adapt to aerobic glycolysis through up-regulation of glucose uptake systems and changes in the expression of metabolic enzymes, which were potential targets for antitumor therapies [1]. It has been identified that the Na^+^-D-glucose co-transporters SGLT1 and SGLT2 are expressed in various tumors, which provides an effective way to attenuate the uptake of glucose and inhibit the growth of tumor cells [2,3,4]. 

Phloridzin, a nonselective SGLT1/SGLT2 inhibitor, was demonstrated to inhibit glucose intestinal absorption and renal resorption, resulting in the normalization of blood glucose and overall reduction of glycaemia in animal models [5,6]. Moreover, phloridzin had been shown a significant in vivo anti-proliferative activity in bladder carcinoma cells that were subcutaneously transplanted from the Fischer 344 male rat’s model, which also inhibited the growth of mammary adenocarcinoma cells subcutaneously transplanted into Fischer 344 female rats [7]. That said, the high cytotoxicity and poor solubility of phlordzin limit its further development for clinical application. 

Trilobatin, an isomer of phloridzin (Figure 1A), was revealed to have lower toxicity and better solubility, also with significant anti-hyperglycemic [8], anti-oxidative [9] and anti-inflammatory properties [10]. It was found in the leaves of *Lithocarpus polystachyus* Rehd. (also called Chinese sweet tea), and had been used as a strong natural sweetener and traditional oriental medicine for many years [10]. Additionally, trilobatin, phloretin (the aglucone of phloridzin) and the other five dihydrochalcone compounds, isolated from the leaves of *Malus* crabapples had been used to test the anti-tumor activity on human cancer cell lines, including A549, Bel7402, HepG2 and HT-29 cell lines, the data indicated that phloretin, 3′″-methoxy-6″-O-feruloy-4′-O-glucopyranosyl-phloretin and 3-hydroxyphloretin had significant effects on the proliferation in all cancer cell lines, but trilobatin had no evident inhibitory activity, especial in HepG2 cells, the IC_50_ is over 300 mM [11].

However, we observed a surprising phenomenon that high doses (over 50 μM) of trilobatin did not show the anti-tumor activity, but selectively induce the proliferation of human hepatoblastoma HepG2. So, in this study, we tried to analyze the interaction between trilobatin and SGLT1/2 through molecular docking, and evaluate the effect of trilobatin on the proliferation of several different cell lines and primary cultured mouse embryonic fibroblasts (**MEF**) and further studied the mechanism involved. These data might provide useful information for further development of trilobatin for therapeutic applications.

## 2. Results

### 2.1. Molecular Docking of SGLT1/2 Inhibitors

To explore the binding mode between SGLT2 and inhibitors, we perform molecular docking between SGLT2 and trilobatin, including trilobatin and its isomer, phloridzin. As shown in Figure 1B, the binding pocket was near residue Tyr290, which was the key residue for receptor and ligand interaction [12]. Both of trilobatin and phloridzin kept the similar pose, especially, the saccharide group had a conformation like glucose transported by SGLT2, which meant the inhibitor could block the interaction between SGLT2 and glucose. The saccharide group of inhibitors had much stronger binding with SGLT2 through hydrogen bonds. In the interaction between trilobatin and SGLT2, there are three strong hydrogen bonds were formed between the hydroxyl groups on saccharide and Asp294/Asp294/Leu71 (~2.9Å, ~2.9Å and ~3.6Å) in SGLT2 respectively, and one more between ether oxygen and Ser74 with ~2.6Å. Moreover, there were three more hydrogen bonds between the hydroxyls on benzene of trilobatin and Ser74/His80/Gly79 with distance ~2.8Å/~3.2Å/~2.8Å, and another hydrophilic interaction between the oxygen and Tyr290/Ser393 (~2.6Å/3.1Å). 

In order to investigate the selectivity of trilobatin for SGLT1/SGLT2, we docked trilobatin into the binding site of SGLT1 by the same way. As shown in Figure 1B, the binding site of SGLT1 and SGLT2 shared many similar residues, especially the pocket composed with the amino acids, including Leu74, Ser77, Tyr290, Asp294 and Gln295 (numbered in SGLT1, these amino acids are labeled as Leu71,Ser74, Tyr290, Asp294 and Gly79 in SGLT2, respectively), which were the key sites for inhibitor binding. Furthermore, the mode of interaction between trilobatin and SGLT1 was similar to SGLT2; there were five hydrogen bonds between the hydroxyls on the saccharide of trilobatin and Asp294/Asp294/Tyr290/Gln295/Ser77 (~2.7Å/~1.7Å/~3.3Å/ ~3.6Å/~2.5Å), and the oxygen could form another 2 hydrogen bonds with Ser77/Tyr290 (~3.0Å/~3.5Å) in hydroxyls on benzene. The results from molecular docking demonstrated that the interaction between trilobatin and SGLT1/2 was similar with phloridzin, hinted that both trilobatin and phloridzin had no selectivity on SGLT1 and SGLT2. 

### 2.2. Effects of Trilobatin, Phloridzin, Canagliflozin, and Empagliflozin on the Cell Proliferation

Mounting references reported that incubation with SGLT inhibitors attenuated the proliferation of tumor cells, and thus showed the prospects of anti-tumor activity [4,13]. In this study, we compared the effects of trilobatin, phloridzin, canagliflozin, empagliflozin on the proliferation of human hepatoblastoma HepG2 cells and Huh 7 cells, the normal human hepatocyte LO2 cells, INS-1 rat pancreatic β cells and primary cultured mouse embryonic fibroblast (MEF) cells. As shown in Figure 2, similar as reported [14], canagliflozin significantly inhibited the proliferation of tumor cells, including HepG2, Huh 7 and INS-1 cells (*p* < 0.01), but it also attenuated the proliferation of the normal human hepatocyte cell line LO2 and primary cultured MEF cells (*p* < 0.01). At the same time, to our surprise, we found that high concentrations of trilobatin did not inhibit, but significantly induced the proliferation of human hepatoblastoma HepG2 (*p* < 0.01). Compared to the control, incubation with 50 or 100 μM trilobatin for 24 h increased the number of viable HepG2 cells by 35% and 50% respectively. 

### 2.3. Effects of Trilobatin, Phloridzin, Canagliflozin, and Empagliflozin on the Cell Cycle and DNA Replication in HepG2 cells

To confirm the effect of these SGLT inhibitors on the proliferation of HepG2 cells, after incubated with indicated concentrations of trilobatin, phloridzin, canagliflozin, and empagliflozin for 24 h, the cell cycle of HepG2 cells was analyzed by flow cytometry. The results suggested that, similar with MTT assay, treatment with 100 μM trilobatin for 24 h induced a markedly decrease of the ratio of G0/G1 phase. Furthermore, we also observed that, treated with 10 μM canagliflozin for 24 h inhibited the cell proliferation, but significantly increased the ratio of G0/G1 phase in HepG2 cells (Figure 3A,B).

Assessment of DNA replication is also an important approach for identifying and quantifying the effect of compounds on cell proliferation [15,16], and 5-ethynyl-2′-deoxyuridine (EDU) had been taken as a powerful tool by incorporating into cellular DNA during DNA replication and accumulating in the nucleus to determine cell cycle kinetics and to disclose potential mitogenic, cytostatic, and cytotoxic effects upon specific cell treatment [17,18,19]. As shown in Figure 3C,D, treatment with 50 μM trilobatin for 24 h induced a significant increase of HepG2 cell proliferation with raising the ratio of EDU-positive cells (*p* < 0.05), but at the same time point, treatment with canagliflozin and empagliflozin markedly inhibited the proliferation of HepG2 cells (*p* < 0.01).

### 2.4. Trilobatin Inhibits Intake of Glucose in Vivo and in Vitro

To explore the mechanism of trilobatin on the proliferation of HepG2 cells, we firstly determined the effect of trilobatin on the intake of glucose in vivo. After 3 g/kg glucose with or without 10 mg/kg trilobatin was administrated (*i.g.*) to C57BL/6 mice, the blood glucose was measured at 0, 15, 30, 60 and 120 min. As shown in Figure 4A,B, trilobatin significantly attenuated the intake of glucose in C57BL/6 mice (*p* < 0.01).

Then, we compared the influence of trilobatin, phloridzin, canagliflozin, and empagliflozin on glucose uptake in HepG2 cells, the results indicated that, similar with the other SGLT inhibitors, trilobatin also attenuated glucose uptake in HepG2 cells, but the efficiency value was much lower than that of others (Figure 4C).

To investigate the role of trilobatin on glucose uptake, we also determined the expression of SGLT2 protein in HepG2, Huh 7, INS-1, LO2 cell lines and primary cultured MEF cells. The results demonstrated that, although SGLT2 was expressed in HepG2 cells, but it is much lower than in INS-1 and LO2 cells (Figure 4D).

### 2.5. Effect of Trilobatin on the Expression of HBXIP and HNF4α in HepG2 Cells

It has been reported that Hepatitis B X-interacting protein (HBXIP) and hepatocyte nuclear factor (HNF)-4α were the two-key regulator of hepatocyte proliferation [20,21,22]. In this study, we firstly determined the effect of trilobatin on the expression of HBXIP and HNF4α. The results from RT-PCR and western blot indicated that trilobatin had no significant role on the mRNA and protein of HBXIP and HNF4α (Figure 5A,B).

We then compared the effects of trilobatin, phloridzin, canagliflozin, and empagliflozin on the protein levels of HBXIP and HNF4α in HepG2 cells, the results demonstrated that incubation with high concentrations of trilobatin for 24 h had no noticeable effects on the protein expression of HBXIP and HNF4α, but canagliflozin, empagliflozin and phloridzin significantly increased the protein level of HNF4α and canagliflozin attenuated the protein levels of HBXIP in HepG2 cells (Figure 5C).

## 3. Materials and Methods

### 3.1. Materials

Trilobatin, phloridzin, canagliflozin, empagliflozin were purchased from Sigma (Sigma, St. Louis, MO, USA). Cell-light EDU Appollo567 in vitro kit was purchased from RIBOBIO (Guangzhou, China). Rat SGLT2 primary antibody was obtained from Cell Signaling Technology (Danvers, MA, USA). HNF4α and *HBXIP* primary antibody was from Proteintech (Rosemont, IL, USA). Specific anti-mouse and anti-rabbit HRP-conjugated second antibodies were obtained from Santa Cruz Biotechnology (Texas, CA, USA). ECL Chemiluminescence Detection Reagent was obtained from Millipore Corporation (Billerica, MA, USA). BCA protein assay kit and other chemicals were purchased from Biyotime (Shanghai, China).

### 3.2. Cell Culture

Human hepatoblastoma HepG2 cells were cultured in a humidified atmosphere containing 5% CO_2_ at 37 °C, and the media were low glucose DMEM medium supplemented with 10% fetal bovine serum (FBS), 100 U/mL penicillin, and 100 μg/mL streptomycin. The medium was changed every other day. 

INS-1 rat pancreatic β cell line, purchased from CCTCC (China Center for Type Culture Collection), was cultured at 37 °C in a humidified atmosphere containing 5% CO_2_. The culture medium was RPMI medium 1640 containing 11 mM glucose and supplemented with 10% FBS, 10 mM HEPES, 100 U/mL penicillin, 100 μg/mL streptomycin, 2 mM L-glutamine, 1 mM sodium pyruvate and 50 μM mercaptoethanol.

Primary cultured mouse embryonic fibrolast (MEF) cells, the normal human hepatocyte cell line LO2 and the *Huh7* hepatocarcinoma *cell* line were cultured in high glucose (25 mM) DMEM medium supplemented with 10% FBS, 100 U/mL penicillin, and 100 μg/mL streptomycin in a water-saturated atmosphere of 5% CO_2_ at 37 °C. 

### 3.3. Molecular Docking of SGLT1/2 Inhibitors

In this study, we used a homology model of human sodium/glucose cotransporter 1 (SGLT1, UniProt entry: P13866) and SGLT2 (UniProt entry: P31639), which were constructed based on the crystal structure of SGLT (PDB entry: 2XQ2, resolution: 2.7 Å) [12]. Briefly, some residues were truncated from the N and C terminals, and Modeller 9.21 [23] was used to construct the homology models by (a) searching and selecting template(s) for the target protein, (b) conducting sequence alignment between the target and template(s), (c) adjusting the sequence alignment, and (d) building and evaluating the homology models.

Next, we adopted the MOLCAD module implemented in SYBYL-X 2.1 to study the potential binding pockets of receptors. The docking program Surflex-Dock GeomX (SFXC) in SYBYL-X 2.1 was used to construct receptor–ligand complexes in which the docking scores were expressed in −log10 (Kd) [24]. The main protocols or parameters of docking were addressed in our previous publications [25,26,27,28]. Briefly, the docking parameters were as follows: (a) the “number of starting conformations per ligand” was set to 10, and the “number of max conformations per fragment” was set to 20; (b) the “maximum number of rotatable bonds per molecule” was set to 100; (c) flags were turned on at “pre-dock minimization”, “post-dock minimization”, “molecule fragmentation”, and “soft grid treatment”; (d) “activate spin alignment method with density of search” was set to 9.0; and (e) the “number of spins per alignment” was set to 12.

### 3.4. Glucose Uptake in Mice

Fourteen to sixteen weeks of C57BL/6 male mice (Chongqing Tengxin Biotechnology Co., Chongqing, China) were allowed ad libitum access to food and water unless otherwise stated, and rooms were maintained at 22 °C and 50% humidity on a 12-h light/dark cycle. For the experiment, 20 male mice of the same age were randomized into two groups, one group was intragastric administrated with 3 g/kg glucose together with 10 mg/kg trilobatin, and another was given with 3 g/kg glucose alone. Blood samples were collected from tail vein at 0, 15, 30, 60 and 120 min respectively, and the blood sugar was determined with One Touch Ultra Mini Blood Glucose Monitoring System (Johnson, Life Scan, Inc., Milpitas, CA, USA). All the experimental protocols were performed in accordance with the principles and guidelines of the Chinese Council Animal Care and also approved by the Institutional Animal Care and Use Committee at Chongqing Science and Technology Committee.

### 3.5. MTT Assay

To measure the effect of trilobatin, phloridzin, canagliflozin, empagliflozin on cell proliferation, after the cells were cultured with indicated concentrations of compounds for different time points, cell viability was determined with 3-(4,5-dimethylthiazol-2-yl)-2,5-diphenyltetrazolium bromide (MTT) colorimetric assay. Generally, after being treated with different doses of SGLT inhibitors for indicated time, the cells were incubated with MTT (0.5 mg/mL final concentration) for 2 h at 37 °C. The formazan crystals were dissolved with DMSO, and the absorbance was measured on a TECAN microplate reader, using a reference wavelength of 630 nm and a test wavelength of 570 nm.

### 3.6. EDU Staining

To confirm the effect of trilobatin on cell proliferation of HepG2 cells, we determined the DNA replication induced by trilobatin in HepG2 cells, which was conducted by using a commercial cell-light EDU Apollo567 in vitro kit (Cat#:C10310-1, RIBOBIO, Guangzhou, China). Briefly, HepG2 cells were seeded in 24-well plates at a cell density of 3 × 10^5^, after cultured overnight, the cells were treated with indicated concentrations of trilobatin, phloridzin, canagliflozin, and empagliflozin for 24 h. Then medium supplemented with EDU (50 μM) was replaced, and continued to culture for 2 h. The cells were washed twice with PBS, stained with 1×Hoechst33342 for 30 min to distinguish the nucleus. 

### 3.7. Cell Cycle Assay

Cell cycle analysis was performed with flow cytometry. Generally, after HepG2 cells were treated with indicated concentrations of trilobatin for 24 h, the cells were trypsinizated and washed twice with phosphate-buffered saline (PBS), and then fixed with methanol. The cells were collected by centrifugation, and resuspended in propidium iodide staining solution (50 μg/mL propidium iodide, 0.05% Triton X-100, 18 μg/mL EDTA and 100 U/mL RNase A in PBS). After incubation for 30 min at room temperature, DNA content was determined by quantitative flow cytometry (Beckman, Coulter, Inc., Middlebury, IN, USA).

### 3.8. RNA Extraction and Quantitative Real-Time PCR

After treated with indicated agents, total RNA was extracted from HepG2 cells using an RNA extraction kit from Bio-Rad (Hercules, CA, USA). Both the quantity and quality of total RNA were analyzed by the TECAN microplate reader. A total of 1 μg of RNA was reverse transcribed with an iScript cDNA synthesis kit. SYBR PCR master mix was used to determine the expression of Txnip on the CFX96 Real-time PCR system (Bio-Rad, Hercules, CA, USA). Primers (β-actin, forward: 5′-GCAAGTGCTTCTAGGCGGAC-3′, reverse: 5′-AAGAAAGGGTGTAAAACGCAGC-3′, HBXIP, forward: 5′- TCGCGTCAAGTGACTGAGG-3′, reverse: 5′- ATG GAGGGATTCTTCATTGTGTC-3′) and HNF4α, forward: 5′- CGAAGGTCAA GCTATGAGGACA-3′, reverse: 5′- ATCTGCGATGCTGGCAATCT-3′) were designed with the open-sourced software Primer 3 Plus (Cambridge, MA, USA). The PCR conditions consisted of 40 cycles with 15 s denaturation at 95 °C, 30 s annealing at 55 °C, and 60 s extensions at 72 °C. The fold change in mRNA was calculated by the 2^-2∆∆Ct^ method using β-actin as the reference gene to normalize the data for all samples.

### 3.9. Western Blot

Cell lysates (20–30 μg) were separated with 10% SDS-PAGE gel and transferred to polyvinylidene difluoride (PVDF) membranes. After being blocked with 5% non-fat milk solution, membranes were probed with specific antibodies (1:2000 dilution) followed by incubation with anti- horseradish–conjugated second antibodies (1:10,000 dilution). Excess antibody was washed off with 20 mM Tris-buffered saline containing Tween-20 (TBST, 150 mM NaCl 20 mM Tris, and 0.1% Tween 20; pH 7.5-7.6). Immunoreactivity was detected using ECL western blotting reagent. Signal bands were analyzed through densitometric scanning using the Quantity One software (Bio-Rad, Hercules, CA, USA).

### 3.10. Statistical Analysis

Results are presented as means ± SD. Analysis of variance was carried out using the software of Origin Lab (Northampton, MA, USA). A one-way ANOVA followed with a post-hoc Tukey’s or Dunnett’s test were used to detect treatment effect and compare the differences among all the groups or selected groups to control. *p* < 0.05 was considered significant.

## 4. Discussion

Glucose is a major metabolic substrate required for growth and survival of cancer cells, and it is mainly imported into cells by glucose transporters (GLUTs). SGLTs are also functionally expressed in various tumor cells, and inhibition of SGLTs blocked glucose uptake and reduced tumor growth and survival, suggesting that SGLT inhibitors, currently in use for treating diabetes, might be useful for cancer therapy [10,29]. Studies have confirmed that inhibition of glucose absorption by SGLT inhibitor attenuated the proliferation of tumor cells [10,29,30]. Recent data showed that dapagliflozin blocked the uptake of the SGLT substrate α-methyl-4-[^18^F] fluoro-D-glucopyranoside (Me4FDG) in tumor xenograph mice models for human pancreatic and prostatic tumors [10], and greatly reduced tumor growth and prolonged survival in autochthonous mouse models and patient-derived xenografts of early-stage lung adenocarcinoma [29].

In the present study, the results from molecular docking showed that trilobatin was a novel SGLT inhibitor, which inhibited the uptake of glucose in SGLT2-expressed HepG2 cells and in C57BL/6 mice. But with surprise that we observed that, different from the other SGLT inhibitors, such as canagliflozin, dapagliflozin, and empagliflozin, high concentrations of trilobatin selectively induced the proliferation of human hepatoblastoma HepG2, but not showed the anti-tumor activity. To confirm the effect of high concentrations of trilobatin on the proliferation of HepG2 cells, we also determined the DNA replication by EDU staining and cell cycle by flow cytometry, all the results clearly showed the proliferative effect of trilobatin in HepG2 cells under high concentration conditions. 

Trilobatin, one of the dihydrochalcones typically present in *Lithocarpus polystachyus* Rehd. was identified as a non-selective inhibitor of SGLT1/2, the property which could be useful in management of postprandial hyperglycemia in diabetes and related disorders. Wang and colleagues demonstrated that leaf extract of *Lithocarpus polystachyus* Rehd. and its major components trilobatin or phloridzin significantly lowered blood glucose, prompted glucose uptake, increased synthesis of liver glycogen, reduced oxidative stress and increased the expression of glucokinase in the liver, suggesting that *Lithocarpus polystachyus* Rehd. leaf extract and its active ingredients trilobatin and phloridzin possessed significant protective activity in diabetic mice [2]. Unfortunately, the results in this study indicated that, although trilobatin had no significant effect on the proliferation of HepG2 cells in the presence of low and moderate concentrations, at high doses, trilobatin actually markedly induced the proliferation of these human hepatocarcinoma cell lines. So, when trilobatin or *Lithocarpus polystachyus* Rehd extracts were administrated to treat diabetes or cancer, the side-effect, especial on liver, should be examined in the future. 

To understand the mechanisms of trilobatin inducing the proliferation of HepG2 cells, we determined the expression of HBXIP and HNF4α, which had been reported to affect the growth of both normal liver cells and hepatoma cells in vitro. Our data demonstrated that treatment with trilobatin had no significant effect on the expression of HBXIP and HNF4α, but another two SGLT inhibitors, canagliflozin and phloridzin significantly increased the protein level of HNF4α and inhibited the proliferation of HepG2 cells respectively, suggesting that HBXIP and HNF4α might be not associated with the effect of trilobatin on the proliferation of HepG2 cells.

## 5. Conclusions

Although phloridzin, an isomer of trilobatin, had been known to have anticancer and antihyperglycaemic properties [5,31], and several other SGLT inhibitors had been proved by FDA as anti-diabetic agents in clinical; in this study, we observed that high concentrations of trilobatin accelerated the proliferation of human hepatoblastoma HepG2 cells, but not in normal human hepatocyte cell line LO2 and other cells. Furthermore, it has been reported that the expression of SGLT2 was increased in liver metastases [32]. All these data hinted that not all the SGLT inhibitors inhibited the proliferation of tumor cells, and further studies are needed to better understand these new glucose-lowering agents in management of cancer and diabetes.

## Figures and Tables

**Figure 1 molecules-24-03390-f001:**
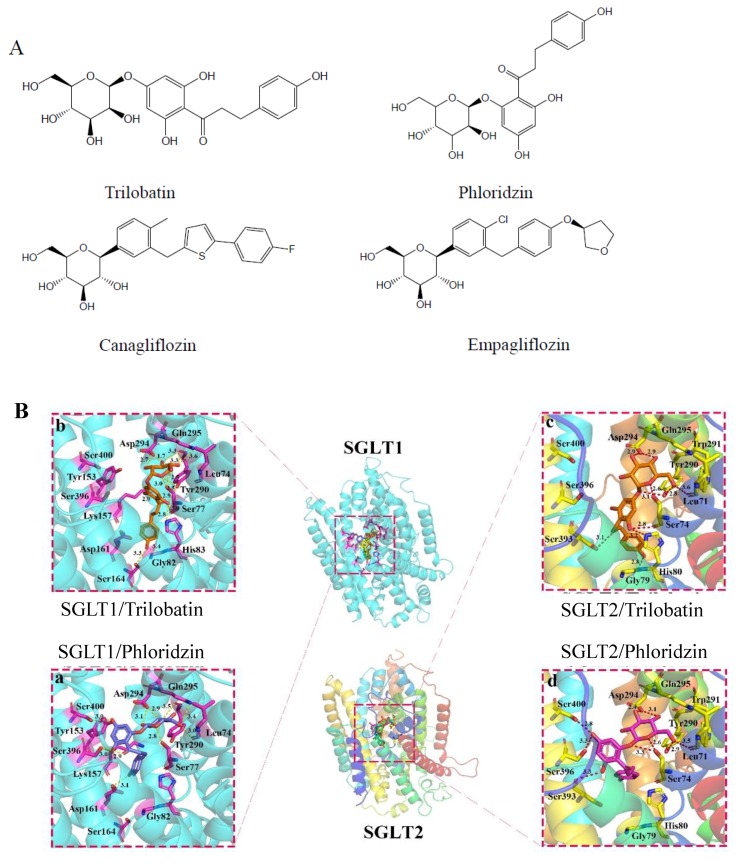
The results of molecular docking. (**A**) The structure of trilobatin, phloridzin, canagliflozin, empagliflozin. (**B**) The interaction between the indicated inhibitors and Na^+^-d-glucose co-transporter (SGLT)1/2.

**Figure 2 molecules-24-03390-f002:**
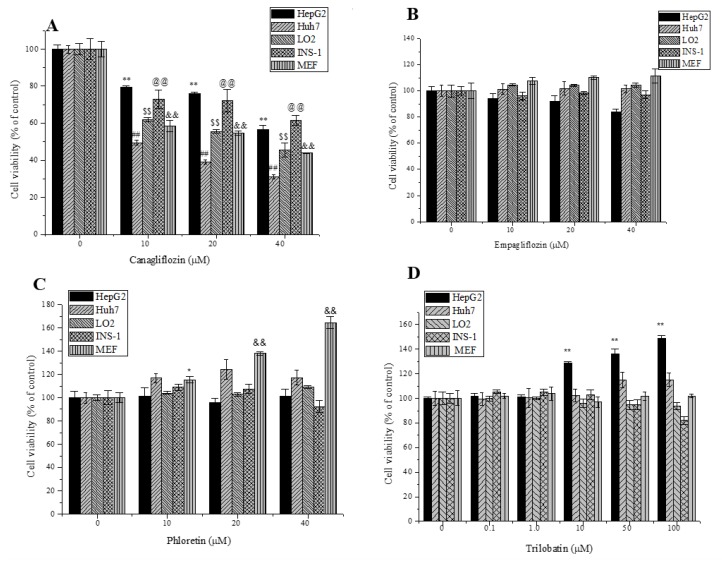
Effects of trilobatin, canagliflozin, empagliflozin and phloridzin on the proliferation of HepG2, Huh 7, LO2, INS-1, and MEF cell lines. After the cells were incubated with indicated concentrations of canagliflozin (**A**), empagliflozin (**B**), phloridzin (**C**), trilobatin (**D**) for 24 h, viable cell number was evaluated by MTT assay, Data are means ± SD (*n* = 12), * *p* < 0.05, _@@_, _##_, ** and _&&_, *p* < 0.01 vs. control.

**Figure 3 molecules-24-03390-f003:**
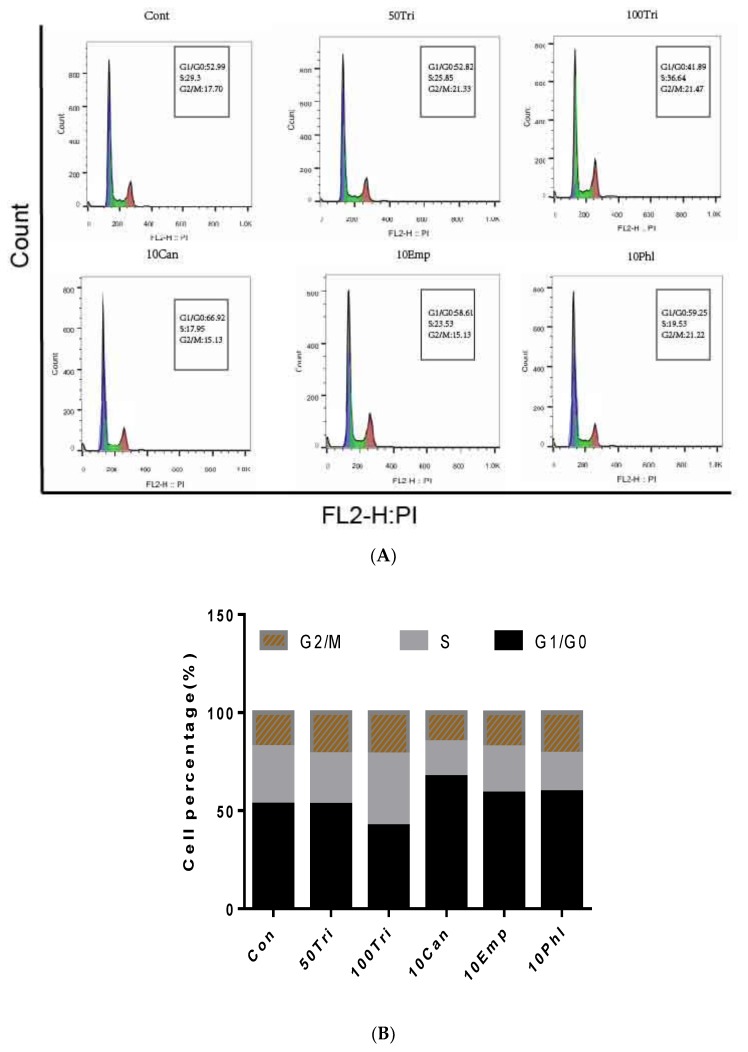
Effects of trilobatin, canagliflozin, empagliflozin and phloridzin on the cell cycle and DNA replication in HepG2 cells. (**A**) After HepG2 cells were incubated with 50 μM trilobatin (50Tri), 100 μM trilobatin (100Tri), 10 μM canagliflozin (10Can), 10 μM empagliflozin (10Emp) and 10 μM phloridzin (10Phl) for 24 h, the cells were collected and cell cycle was measured by flow cytometry. (**B**) The statistical results of flow cytometry assay for cell cycle after treated with rilobatin, canagliflozin, empagliflozin and phloridzin respectively. (**C**) Proliferating HepG2 cells were labeled with EDU, the Click-it reaction showed EDU staining (red), and cell nuclei were stained with Hoechst 33342 (blue). The images are representative of the obtained results (Scale bar = 100 nM). (**D**) The percentage of EDU-positive HepG2 cells was calculated from three random fields, * *p* < 0.05, ** *p* < 0.01 vs. control (Con).

**Figure 4 molecules-24-03390-f004:**
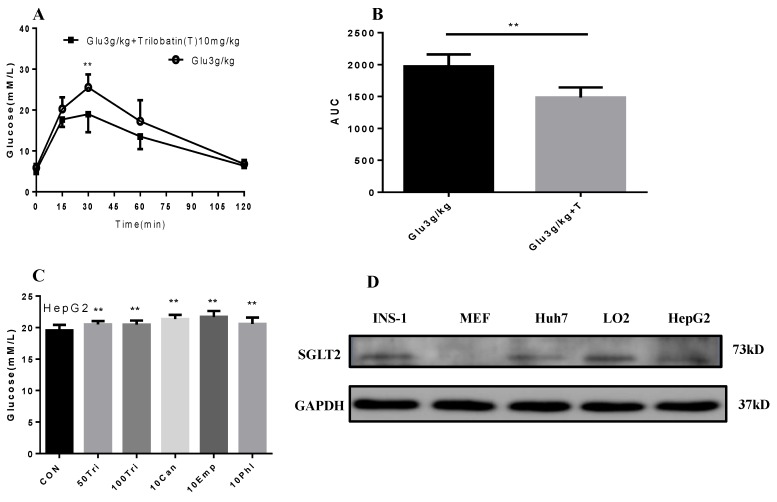
Trilobatin suppressed the intake of glucose in vivo and in vitro. (**A**) After 3 mg/kg glucose was intragastric administrated with or without 10 mg/kg trilobatin, blood glucose was measured at 0, 15, 30, 60 and 120 min. (**A**) the contents of blood glucose at 0, 15, 30, 60 and 120 min, Data are means ± SD (*n* = 8), ***p* < 0.01 vs. control. (**B**) The area under curve (AUC) of Figure 3A. (**C**) After HepG2 cells were replaced onto 24-well plate and cultured overnight, the cells were washed once with phosphate-buffered saline (PBS) and starved 2 h, and then the media was changed with high glucose (25 mM) completed media, and continued to incubate for 2 h, the supernatant was collected to determine the content of glucose according to the protocol from supplier. Data are means ± SD (*n* = 12), ***p* < 0.01 vs. control. (**D**) The protein levels of SGLT2 in INS-1, MEF, Huh7, LO2 and HepG2 cells.

**Figure 5 molecules-24-03390-f005:**
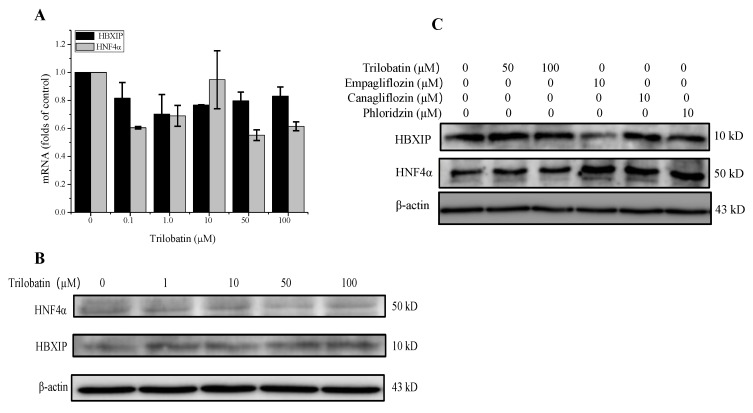
Effects of trilobatin on the expression of HBXIP and HNF4α in HepG2 cells. (**A**) After HepG2 cells were treated with indicated dose of trilobatin for 24 h, the mRNA of HBXIP and HNF4α were measured by RT-PCR, Data are mean ± SD (*n* = 3). (**B**) After HepG2 cells were incubated with indicated concentrations of trilobatin for 24 h, the protein levels of HBXIP and HNF4α were detected with western blot. (**C**) After HepG2 cells were incubated with indicated concentrations of trilobatin, canagliflozin, empagliflozin and phloridzin for 24 h, the cell lysates were used to determine the protein levels of HBXIP and HNF4α by western blot.

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
