# Peer review of "Trilobatin, a Novel SGLT1/2 Inhibitor, Selectively Induces the Proliferation of Human Hepatoblastoma Cells"

_molecules, 2019, doi:10.3390/molecules24183390_

Round 1

Reviewer 1 Report

The manuscript entitled “Trilobatin, a novel SGLT1/2 inhibitor, selectively induces the proliferation of human hepatoblastoma cells” identified trilobatin as a new SGLT1/2 inhibitor by docking simulation and biochemical analyses on in vitro and in vivo. The trilobatin showed cell type selective toxicities in vitro. On the other hand, proliferation of the HepG2 cells was induced by treatment of the trilobatin.

The study includes important information of the action of SGLT inhibitors to cancer cells and inhibitory activity of hyperglycemia. The authors carefully described the results and discussion.

Comments:

1). In Figure 2, trilobatin induced proliferation of HepG2 cells, but not in case of MEF. On the other hand, phlorizin, a SGLT-1 inhibitor, did not induced proliferation of HepG2 cells, but induced that of MEF. How do the authors explain? These compounds are estimated as SGLT-1 inhibitor by docking simulation. What is the difference to the action of these compounds?

2). Trilobatin induced proliferation of HepG2 cells, but arrested the cell cycle at S phase. How does the cell cycle arrest affect to the induction of the cell proliferation?

Reviewer 2 Report

In the original paper: “Trilobatin, a novel SGLT1/2 inhibitor, selectively induces the proliferation of human hepatoblastoma cells”, the molecular docking, and the effect of trilobatin on the cell proliferation were performed.

All the experiments and studies are sound and well done.
The science in the paper is strong and well conducted. In my opinion the manuscript is written carefully and it provides interesting results. In summary, I recommend this manuscript for publications after the stated minor corrections.

Additional suggestions:

1. At what dose trilobatin inhibits SGLT. How does this dose relate to the dose that produces the effect of cancer cell growth? 2. Literature 18 should be improved 3. In figure 1 correct the name phlorizin (phloridzin)

Author Response

something wrong with the website.

Reviewer 3 Report

The article is written in a nice language, it is concise and interesting. The methodology used is correct and the results obtained are very interesting, given the fact that they are quite surprising. They certainly require further in-depth research. This is an interesting voice in the discussion about trilobatin. I recommend publishing it in its current form, after minor editorial corrections.

Author Response

Reviewer 3

Comments and Suggestions for Authors

The article is written in a nice language, it is concise and interesting. The methodology used is correct and the results obtained are very interesting, given the fact that they are quite surprising. They certainly require further in-depth research. This is an interesting voice in the discussion about trilobatin. I recommend publishing it in its current form, after minor editorial corrections.

Response: Thank you for your overprize. Although we observed an interesting phenomena during our research, but there are some questions to be clarified in future.

Round 2

Reviewer 1 Report

The manuscript was improved. The present manuscript will contribute to reveal the effects of the SGLT-1 inhibitors against proliferation of cancer cells. This study is a first step to solve the action of the SGLT-1 inhibitors.